# Multitask Transformer Models for Demographic and Industry Profiling on Long-Form Blog Texts

**Bahor Eshmirzayeva**                                          *b.eshmirzayeva@tsue.uz*
*Department of Economics*
*Tashkent State University of Economics, Tashkent, Uzbekistan*

**Shirali Kadyrov**                                          *shirali.kadyrov@sdu.edu.kz*
*School of Information Technologies and Applied Mathematics*
*SDU University, Kaskelen, Kazakhstan*

**Reviewed on OpenReview:** *https://openreview.net/forum?id=WtFwcCvt9i*

## Abstract

We address the challenge of multitask author profiling on long-form blog text, where standard Transformer models are limited by fixed input length constraints that prevent full utilization of extended documents. To overcome this, we develop four transformer-based models that jointly predict gender, age group, and industry. Using a cleaned version of the Blog Authorship Corpus, we explore document-length handling strategies that span input ranges from 192 to 500 tokens, including long-context encoding, BART-based summarization, and chunked processing with prediction fusion. Our experiments show that multitask learning consistently outperforms strong single-task baselines, with the largest gains for industry. We further find that broader input context yields more reliable predictions, while alternative representations emphasize complementary stylistic and topical cues. Taken together, these findings provide a comprehensive analysis of text-length effects in multitask author profiling and highlight the importance of contextual breadth for robust demographic inference. The dataset was preprocessed by merging industry tags into fourteen categories and applying standard text normalization.

## 1 Introduction

Author profiling is the task of inferring demographic traits of an individual — including gender, age, occupation, and personality — from their written text (Rangel et al., 2018). The task has practical uses in areas like forensic analysis, social-behavior research, improving personalised recommendations, and identifying misinformation, deception, or harmful accounts on social media (Ott et al., 2011; Mishra et al., 2019; Lanza-Cruz et al., 2023). Extended personal writing, such as blog posts, remains a particularly rich source of signal because it simultaneously exposes linguistic style, topical preferences, and sociolectal patterns that correlate strongly with real-world author attributes (Schler et al., 2006; Nguyen et al., 2016).

The ability to automatically infer demographic attributes from text has far-reaching implications across multiple domains. In cybersecurity and platform integrity, demographic inference enables detection of coordinated inauthentic behavior, sockpuppet accounts, and age-misrepresenting users on social platforms. In forensic linguistics, it supports authorship analysis in legal investigations. In personalization systems, understanding user demographics improves recommendation quality and content targeting. Beyond these applications, demographic inference from text raises important questions about privacy — recent work has demonstrated that large language models can infer sensitive personal attributes from text with alarming accuracy Staab et al. (2024), underscoring the need to understand what demographic signals text actually contains and how they can be modeled.

Despite the rich signal available in long-form blog text, existing author profiling research has not fully exploited it. Prior multitask approaches rely on pre-transformer architectures that struggles with extended context, while transformer-based studies typically focus on individual demographic traits or overlook industry sector prediction entirely. Furthermore, no existing work has systematically investigated how different long-document handling strategies affect demographic inference across multiple tasks simultaneously. This leaves a critical gap: there is no transformer-based multitask framework that jointly predicts gender, age group, and industry sector from long-form blog posts — a gap the present study directly addresses.

The main contribution of our approach is threefold: (1) We present the first transformer-based multitask framework that jointly predicts gender, age group, and a fine-grained 14-class industry sector from real blog posts, using the cleaned Blog Authorship Corpus (Eshmirzayeva, 2025b). (2) We conduct a controlled comparison of four long-document representation strategies—truncation, extended-context encoding, summarization-based compression, and chunk-based processing—and quantify their differential impact across all three profiling tasks. (3) We uncover a key novel finding: gender and age-group prediction are notably robust to aggressive truncation, summarization, and chunking, whereas industry classification suffers sharp performance degradation under the same conditions. This indicates that professional cues are far sparser and more widely distributed throughout the document than demographic signals, with important implications for long-document author profiling systems.

The rest of the paper is organized as follows. Section 2 reviews related work. Section 3 describes the dataset, preprocessing pipeline, and proposed models. Section 4 presents experimental results, Section 5 discusses findings and Section 6 concludes the paper.

## 2 Related Work

Language use systematically varies with social factors such as age, gender, and occupation — a finding well established in computational sociolinguistics that provides the theoretical foundation for author profiling Nguyen et al. (2016). Modern NLP architectures have given this foundation powerful tools to build on. Transformer-based models Vaswani et al. (2017) introduced self-attention for sequence modeling, with BERT Devlin et al. (2019) demonstrating the effectiveness of bidirectional pre-training for classification, later refined by models such as RoBERTa Liu et al. (2019) and DeBERTa He et al. (2021). These architectures brought the representational depth that demographic inference from text requires. More recent work has continued in this direction, applying transformer-based models augmented with handcrafted stylistic features to PAN shared-task benchmarks, demonstrating that purely attention-based representations can be further strengthened by explicit stylistic signals López-Santillán et al. (2023). Despite this progress, a comprehensive survey HaCohen-Kerner (2022) confirms that even after years of competition across PAN shared tasks, age prediction accuracies remain surprisingly low, and occupation inference from unstructured personal writing remains unexplored.

Multitask learning has been a natural fit for author profiling since demographic attributes — gender, age, and occupation — share overlapping lexical, stylistic, and topical cues that single-task models leave partially unexploited. Early benchmarks in author profiling, such as the PAN 2016 shared task Rangel et al. (2016), highlighted the difficulty of cross-genre generalization. Their findings demonstrated that while models trained on Twitter could predict gender with reasonable accuracy, their performance on blogs and reviews was heavily mediated by document length and genre-specific stylistic norms. On the Blog Authorship Corpus specifically, prior work Jiang et al. (2018) explored multitask hierarchical representations using CNN and LSTM architectures for gender, age, and industry prediction — the closest prior work to ours, yet relying on pre-transformer architectures whose representational capacity is more limited than modern encoders. Transformer-based work on the same corpus followed, with BERT applied Thakur & Tickoo (2023) to predict age and gender from blog posts, achieving strong results yet training each task independently and leaving industry untouched. Beyond the Blog Authorship Corpus, prior work Abdul-Mageed et al. (2019) demonstrated that sentence-level BERT in a multitask setting effectively captures joint gender and age signals from social media text, demonstrating the effectiveness of combining transformers with multitask learning. More recently, Türkmen and Kutlu (2025) demonstrated that message ordering and content selection strategies affect age, gender, and occupation detection performance on social media Türkmen & Kutlu (2025). They

found that shuffling messages improves model robustness to adversarial attacks and reduces the impact of noisy or irrelevant content on short social-media posts. Broader benchmarks such as the Italian TAG-it task extended the multitask framing to age, gender, and topic prediction on forum posts in Italian, finding that topic and gender are easier to predict than age, and that fine-tuned BERT-based models outperformed classical approaches in both subtasks where topic labels represent thematic interests Cimino et al. (2020). That occupation-level inference from personal writing is at least feasible was shown in the PAN celebrity profiling task Wiegmann et al. (2019), where professional categories were predicted from Twitter timelines — yet this remained a single-task, short-text setting. Further evidence that professional signal is latently present in writing comes from work on contrastive pretraining Huertas-Tato et al. (2025), which showed that training on diverse text, including blogs, produces authorship embeddings that naturally reflect gender, age, and occupation, even without explicit supervision. Multitask fine-tuning itself is not without pitfalls: shared training can have inconsistent effects on underrepresented groups, motivating careful design of shared representations and loss weighting Kulkarni et al. (2024) — considerations that we aim to address in our architecture. However, joint age-gender-occupation prediction using modern encoders on long-form personal writing remains unexamined, despite evidence that all three signals are present in such text.

Processing long documents with transformer-based models presents a well-recognized challenge. Architectures such as Longformer and BigBird addressed the quadratic complexity of full self-attention by introducing sparse and local attention patterns, extending the practical context window to thousands of tokens and enabling document-level classification at scale Beltagy et al. (2020); Zaheer et al. (2020). These contributions demonstrated that architectural innovations can substantially widen the context a model processes. However, a growing body of evidence shows that a wider context window does not automatically translate into effective use of that context. Manakul and Gales demonstrate that standard Transformer models tend to prioritize local context over global document structure, leading to alternative strategies such as content selection and local attention for long-span summarization Manakul & Gales (2021). Liu et al. identify a positional bias they term "lost in the middle": performance systematically degrades when relevant information appears in the middle of long sequences, indicating that models rely on positional heuristics rather than genuinely integrating the full context Liu et al. (2023). Gao et al. further show a discrepancy between internal representations and output behavior — models may encode relevant long-range information yet consistently fail to utilize it during generation or classification Gao et al. (2024). Benchmarks specifically designed for long-context evaluation, such as LooGLE and ETHIC, confirm that state-of-the-art models often fail to capture extended dependencies and rely on only partial context even when comprehensive coverage is required Li et al. (2024); Lee et al. (2025). These findings suggest that the challenge in long-document author profiling is not simply one of context length, but of how effectively distributed signals can be extracted and aggregated across extended text. This is addressed by systematically comparing four input strategies, rather than relying on a single architectural solution.

Our work focuses on specialized encoder models rather than large language models (LLMs) for demographic classification. For structured classification tasks where labeled data is available, fine-tuned encoders offer a compelling balance of performance, computational efficiency, and interpretability. Recent evidence supports this design choice: Zhang et al. (2025) demonstrate that BERT-like encoder models frequently outperform LLMs on classification tasks characterized by surface patterns and sufficient training data Zhang et al. (2025). Nevertheless, LLMs have also shown promise in demographic inference tasks. Cho et al. (2024) evaluate LLMs for author profiling of age and gender from textual data, demonstrating that these models capture demographic signals effectively in independent single-task settings Cho et al. (2024). Oulahbib et al. (2026) further propose Deep-AP, an efficient multitask architecture built on LLaMA-3.2 (among other backbones) that jointly performs author-level age and gender profiling together with post-level depression detection, showing gains in both accuracy and computational efficiency Oulahbib et al. (2026). Huang et al. (2024) show that LLMs encode meaningful stylistic patterns relevant to authorship tasks Huang et al. (2024), while Sanku (2024) explores zero-shot prompting and fine-tuning strategies for gender prediction Sanku (2024). Staab et al. (2023) further demonstrate that LLMs can infer sensitive personal attributes—including demographic information—from natural language text at inference time, even when such information is not explicitly stated, confirming that linguistic cues carry implicit demographic signals Staab et al. (2024). While these results demonstrate LLM capabilities, they also confirm that for our setting—multitask demo-

graphic classification on long-form personal writing with available labels—specialized encoders remain the appropriate architectural choice.

Three critical questions remain unaddressed: (1) how modern transformer encoders compare for joint age-gender-occupation prediction on long blogs, (2) which input strategies effectively capture professional cues across lengthy text, and (3) whether multitask learning helps or hinders occupation inference when combined with age and gender prediction. We address these questions through a systematic evaluation of four document-length handling strategies under a joint multitask learning framework. Our work is the first to systematically demonstrate that long contiguous sequences are critical for industry inference while gender and age remain robust under truncation or summarization, resulting in improved performance across the tasks.

## 3 Methodology

This section describes the dataset cleaning and preprocessing pipeline, the single-task baselines, and our four multitask transformer models with different document representation strategies.

### 3.1 Dataset Preprocessing

We use the Blog Authorship Corpus, a collection of individual blog posts written before 2004 (Tatman, 2017), and rely on the cleaned and standardized version introduced in (Eshmirzayeva, 2025b). In total, our cleaned dataset includes 96,199 posts and 43 million words authored by 5,477 bloggers—averaging 17.6 posts and approximately 7,900 words per author Table 1. The dataset is refined for author profiling and NLP tasks by applying the preprocessing steps summarized in Fig. 1, which include text cleaning and normalization, merging semantically related industry labels, and splitting long posts into 200–1,500-word chunks. Stopwords and emojis are retained to preserve stylistic and affective cues and to maintain the contextual integrity required by transformer-based models. Summaries are generated using the BART-Large-CNN model for efficient downstream processing (Lewis et al., 2020). Table 2 summarizes the final distributions of gender, age, and industry categories retained in the dataset. Code and configurations are available at (Eshmirzayeva, 2025a).

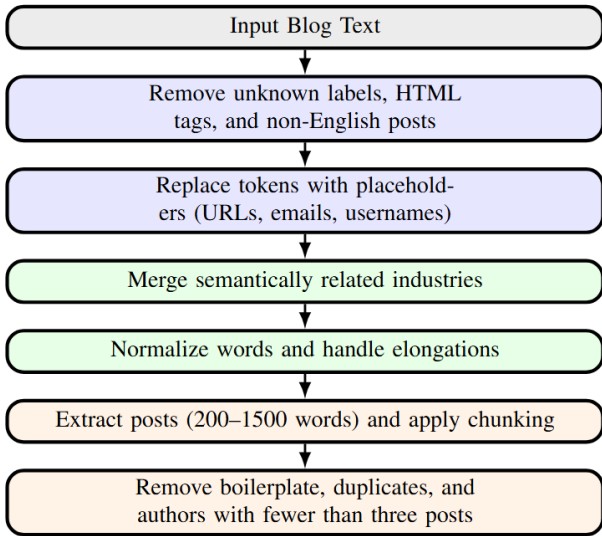

Figure 1: Dataset preprocessing pipeline. Blue = cleaning, green = normalization, orange = filter- ing.

Table 1: Author and Blog Post Statistics

| Metric | Value |
|---|---|
| Unique authors | 5,477 |
| Total blog posts | 96,199 |
| Average posts per author | 17.56 |
| Median posts per author | 7 |
| Minimum posts per author | 3 |
| Maximum posts per author | 584 |

Table 2: Distribution of Gender, Age Group, and Industry in the dataset.

| Attribute | Category | Count | % |
|---|---|---|---|
| Gender | Male | 51,002 | 53.02 |
| | Female | 45,197 | 46.98 |
| Age Group | 13–17 | 17,471 | 18.16 |
| | 18–29 | 55,377 | 57.57 |
| | 30–48 | 23,351 | 24.27 |
| Industry | Student | 10,695 | 11.12 |
| | Education | 10,322 | 10.73 |
| | Arts | 9,728 | 10.11 |
| | Technology | 9,560 | 9.94 |
| | Creative Media & Culture | 8,327 | 8.66 |
| | Science & Technical | 7,371 | 7.66 |
| | Communications-Media | 6,760 | 7.03 |
| | Business & Cons. | 6,371 | 6.62 |
| | Industrial & Misc. | 5,901 | 6.13 |
| | Law & Spec. Services | 5,187 | 5.39 |
| | Internet | 4,641 | 4.82 |
| | Non-Profit | 4,629 | 4.81 |
| | Public Service & Gov. | 4,067 | 4.23 |
| | Finance & Property | 2,640 | 2.74 |

## 3.2 Model Architectures and Training

We first introduce the single-task baselines used for comparison. Logistic Regression, SVM, LSTM, and BERT are trained independently to predict gender, three age groups, and fourteen industries using the cleaned Blog Authorship Corpus. To investigate the benefits of joint learning and extended context, we introduce four multitask learning (MTL) transformer variants that share a single encoder followed by three task-specific classification heads.

The variants are designed to systematically compare input processing strategies within an otherwise identical MTL framework:

RoBERTa-192: RoBERTa-base processing 192-tokens of each post.

DeBERTa-500: DeBERTa-v3-base taking full sequences up to 500 tokens.

DeBERTa-Summary: DeBERTa-v3-base applied to single BART-large-CNN summaries truncated to 256 tokens.

DeBERTa chunking-model that segments long posts to capture full-document information.

Single-task baselines are trained using class-weighted cross-entropy loss for each task independently, ensuring that minority classes were fairly represented during optimization. Figure 2 summarizes the shared-encoder, multi-head setup and the input-handling variants. For the multitask (MTL) models, the total loss is the weighted sum of the individual task losses:

$$\mathcal{L} = \mathcal{L}_g + \mathcal{L}_a + 1.5\,\mathcal{L}_i$$

We apply a weight of 1.5 to the industry loss to compensate for its higher difficulty and cardinality. All models are optimized with AdamW (weight decay 0.01) using a linear learning-rate schedule with 10% warmup steps and learning rate 2e-5 in Table 3. The dataset is split into 85% training and 15% test sets,

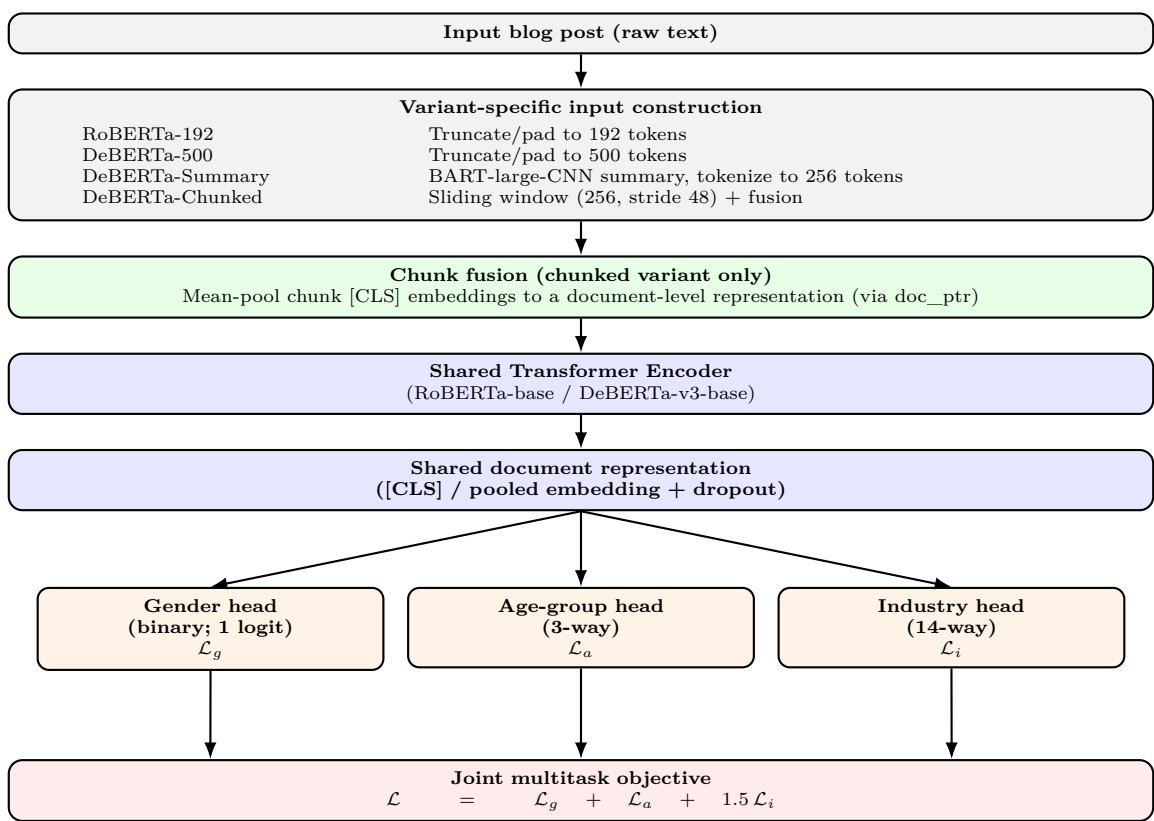

Figure 2: MTL architecture with *parallel* task heads and hard parameter sharing. Model variants differ only in input construction (truncation, long-context encoding, summarization, or chunking with mean pooling), while all heads are trained jointly with a weighted loss.

Table 3: Model Configuration and Training Hyperparameters

| Hyperparameter | Value |
|---|---|
| Batch size (train) | 4, 16, 32 |
| Batch size (validation) | 4, 16, 32 |
| Sequence length | 192, 256, 500 |
| Learning rate | $2 \times 10^{-5}$ |
| Dropout | 0.1 |
| Weight decay | 0.01 |
| Warmup steps | 100 |
| Optimizer | AdamW |
| Scheduler | Cosine decay with warmup |

stratified by industry to preserve distributions. We use different batch sizes across models depending on their computational cost. To prevent data leakage, we use an author-level train–test split: all posts from each author are assigned exclusively to either the training set or the test set. This guarantees evaluation on entirely unseen writers. Due to the high computational cost, we report results based on a single training run using a fixed random seed of 42 for reproducibility. All models are evaluated using macro-averaged F1-score and accuracy, with MTL models using a composite F1-score (average across tasks). Training is performed using PyTorch on cloud-based GPUs provided by Kaggle and Google Colab (T4/A100).

### 3.2.1 Single-Task Machine Learning Baselines

For comparison, we trained classical machine learning baselines (Logistic Regression and SVM) using TF-IDF features, and deep learning models (BERT and LSTM) on raw text sequences under the same split. The TF-IDF vectorizer was configured with unigrams and bigrams to capture both individual word and short phrase patterns, excluding tokens appearing in fewer than two documents or in more than 90% of documents.

The BERT-base-uncased single-task baseline is fine-tuned for sequence classification with a maximum input length of 192 tokens, ensuring direct comparability with the RoBERTa-192 MTL variant. The bidirectional LSTM likewise operates on sequences truncated or padded to 192 tokens. Both models follow the same training protocol as the MTL transformers. The LSTM required more training epochs because we kept the learning rate consistent with the transformer models, which slows convergence but ensures a fair and controlled comparison.

### 3.2.2 RoBERTa-Based MTL with 192-Token Length

Our first MTL model uses RoBERTa-base to jointly predict gender, age group, and industry, leveraging shared linguistic features. The novel combination of these tasks, tailored for the cleaned Blog Authorship Corpus, uses a shared RoBERTa encoder (768-dimensional [CLS] token output) with dropout (p=0.1) and task specific linear heads for gender (R1), age group (R3), and industry (R14). Lg is binary cross-entropy for gender, and La and Li are cross-entropy losses with label smoothing (0.05) for age group and industry, with 1.5 weight on industry to prioritize its complexity. Texts are tokenized (max length: 192) and dynamically padded. The model was fine-tuned for 10 epochs (batch size: 16) with AdamW, a cosine scheduler (100 warmup steps), and gradient clipping (1.0).

### 3.2.3 DeBERTa-Based MTL with 500-Token Length

To capture longer contexts, we developed an MTL model using DeBERTa-V3-base, novel for its disentangled attention mechanism. It predicts the same tasks as the RoBERTa model, using a shared encoder (768-dimensional [CLS] output) with dropout (p=0.1) and identical task-specific heads. Texts are tokenized with a maximum length of 500 and dynamically padded. The model is fine-tuned for 16 epochs using the same architecture as the other variants. Model checkpointing saved the best and last states based on composite F1-score, enhancing training robustness.

### 3.2.4 MTL with Summarized Text and 256-Token Length

To explore concise inputs, we apply our DeBERTa-V3-based MTL model to summarized texts generated by facebook/bart-large-cnn, a novel approach to reduce text length while preserving key information. Blog posts are summarized using beam search (beam size: 5) with length constraints and repetition mitigation. Summaries are tokenized with a maximum length of 256 and dynamically padded. The model architecture mirrors the RoBERTa model, fine-tuned for 12 epochs with identical hyperparameters and checkpointing.

### 3.2.5 MTL with Chunked Text

To handle long posts, we develop a chunking-based variant of our DeBERTa-V3-base MTL model. While it retains the same underlying architecture and optimization setup as the original model, we introduce a sliding-window tokenizer (stride = 48, max length = 256) to segment extended posts into overlapping chunks. Chunk embeddings are mean-pooled (attention-masked), passed to task-specific heads, and the resulting logits are averaged per document to compute document-level losses and predictions. The model is fine-tuned for 19 epochs. This approach enhances feature extraction from longer texts while remaining compatible with the existing MTL framework.

### 3.3 Evaluation Metrics

Let $\mathcal{D} = \{(x_n, y_n)\}_{n=1}^{N}$ be the test set for a given task with $C$ classes, where $y_n \in \{1, \ldots, C\}$ is the gold label and $\hat{y}_n$ is the predicted label. We report *Accuracy* and *Macro-F1* for each task, and a *Composite Macro-F1* that aggregates performance across the three tasks (gender, age group, industry).

**Accuracy.** Accuracy is the fraction of correct predictions:

$$\text{Acc} = \frac{1}{N} \sum_{n=1}^{N} \mathbb{I}[\hat{y}_n = y_n], \tag{1}$$

where $\mathbb{I}[\cdot]$ is the indicator function.

**Per-class Precision, Recall, and F1.** For class $c$, define true positives, false positives, and false negatives as

$$\text{TP}_c = \sum_{n=1}^{N} \mathbb{I}[y_n = c \wedge \hat{y}_n = c], \quad \text{FP}_c = \sum_{n=1}^{N} \mathbb{I}[y_n \neq c \wedge \hat{y}_n = c], \quad \text{FN}_c = \sum_{n=1}^{N} \mathbb{I}[y_n = c \wedge \hat{y}_n \neq c]. \tag{2}$$

Precision and recall for class $c$ are:

$$P_c = \frac{\text{TP}_c}{\text{TP}_c + \text{FP}_c}, \qquad R_c = \frac{\text{TP}_c}{\text{TP}_c + \text{FN}_c}, \tag{3}$$

and the class-wise F1-score is:

$$F1_c = \frac{2P_c R_c}{P_c + R_c}. \tag{4}$$

(When a denominator is zero, we follow the standard convention of setting the corresponding quantity to zero.)

**Macro-F1.** Macro-F1 averages the class-wise F1-scores uniformly, giving equal weight to each class:

$$\text{MacroF1} = \frac{1}{C} \sum_{c=1}^{C} F1_c. \tag{5}$$

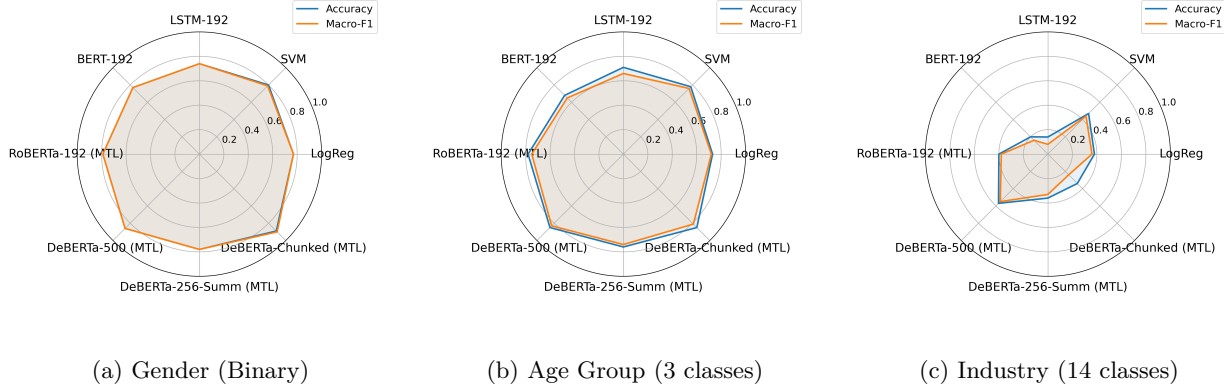

(a) Gender (Binary)      (b) Age Group (3 classes)      (c) Industry (14 classes)

Figure 3: Radar plots compare Accuracy and Macro-F1 across single-task and multitask models.

**Composite Macro-F1 across tasks.** Let $\text{MacroF1}_g$, $\text{MacroF1}_a$, and $\text{MacroF1}_i$ denote macro-F1 for gender, age group, and industry, respectively. We define the composite score as:

$$\text{CompositeF1} = \frac{\text{MacroF1}_g + \text{MacroF1}_a + \text{MacroF1}_i}{3}. \tag{6}$$

This metric summarizes overall multitask performance while treating the three tasks equally.

## 4 Results of Experiments

Table 4 reports accuracy and macro-F1 scores for all baselines and multitask models. Single-task baselines (Logistic Regression, SVM with TF-IDF, LSTM, and BERT) reach 0.74–0.79 macro-F1 on gender, 0.65–0.76 on age group, and 0.08–0.44 on the 14-class industry task. Classical single-task models perform reasonably on gender and age but struggle dramatically on the 14-class industry task, with the strongest baseline (SVM) reaching only 0.47 accuracy and 0.44 macro-F1. This confirms that industry prediction requires richer contextual representations than those captured by bag-of-n-grams or short-context models. All four multitask transformer variants substantially outperform the single-task baselines.

Joint multitask training improves performance on all three tasks compared to training separate single-task models on the same architecture. The DeBERTa-V3-base model with a 500-token context is the best-

Table 4: Performance of Single-Task ML and Multitask Learning Models on Gender, Age Group, and Industry Prediction Tasks

| Model | Epochs | Gender Acc / F1 | Age Group Acc / F1 | Industry Acc / F1 | Composite F1 |
|---|---|---|---|---|---|
| *Single-Task ML Baselines* | | | | | |
| Logistic Regression | | 0.77 / 0.77 | 0.73 / 0.72 | 0.38 / 0.36 | – |
| SVM | | 0.80 / 0.79 | 0.78 / 0.76 | 0.47 / 0.44 | – |
| LSTM (192 tokens) | 30 / 66 / 26 | 0.74 / 0.74 | 0.71 / 0.66 | 0.14 / 0.08 | – |
| BERT (192 tokens) | 4 / 4 / 2 | 0.77 / 0.77 | 0.68 / 0.65 | 0.20 / 0.16 | – |
| *Multitask Learning Models* | | | | | |
| RoBERTa (192 tokens) | 10 | 0.79 / 0.79 | 0.78 / 0.75 | 0.40 / 0.38 | 0.64 |
| DeBERTa (500 tokens) | 16 | 0.86 / 0.86 | **0.85 / 0.83** | **0.57 / 0.55** | **0.75** |
| DeBERTa (256 tokens, Summ.) | 12 | 0.78 / 0.78 | 0.76 / 0.74 | 0.36 / 0.33 | 0.62 |
| DeBERTa (chunked) | 19 | **0.89 / 0.90** | **0.85** / 0.81 | 0.34 / 0.24 | 0.65 |

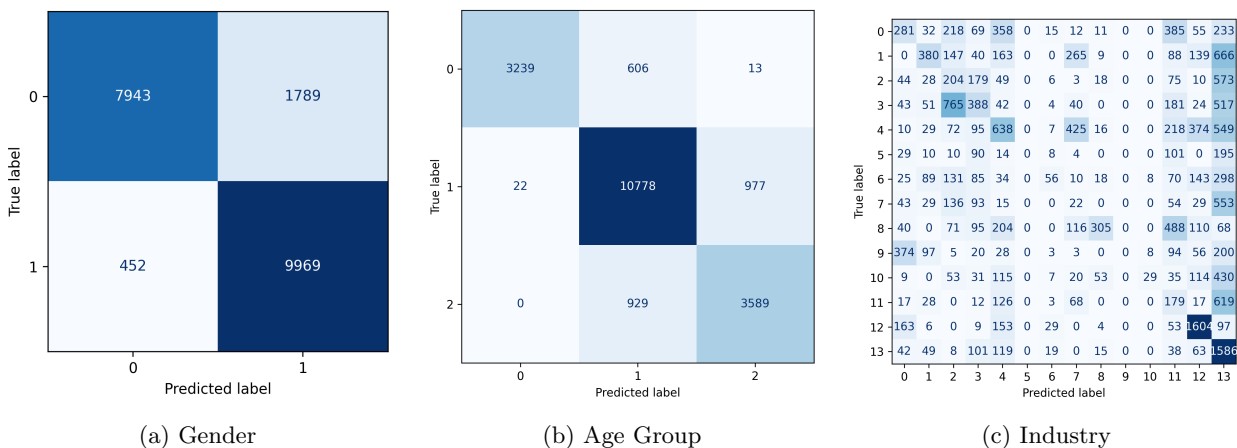

Figure 4: Confusion matrices (absolute counts) for the model overall (DeBERTa (Chunked)).

performing configuration, achieving the highest composite macro-F1 (0.75) across all setups, along with strong per-task gains: gender F1 of 0.86, age-group F1 of 0.83, and, most notably, industry F1 of 0.55 (+11 points over the best single-task model). Extending the context length from 192 tokens (RoBERTa) to 500 tokens (DeBERTa) further increases scores on every task. Replacing the contiguous 500-token input with either BART summarization or sliding-window mean pooling preserves most of the performance on gender and age group while reducing industry macro-F1 substantially. Figure 3 summarizes these trends using radar plots, which compare Accuracy and macro-F1 across models for each task. The plots highlight the consistent gains from multitask learning and make clear that improvements are largest and most uneven for industry prediction, while gender and age exhibit more stable performance across modeling choices. Although the sliding-window variant achieves the strongest gender macro-F1 (0.90) among the multitask models, its very low industry macro-F1 (0.24) substantially reduces its overall composite performance. This contrast shows that gender and age cues remain fully recoverable when the input document is split into short independent chunks, whereas industry prediction requires coherent long-range context that is disrupted by chunking. These results confirm that long contiguous context is the decisive factor for strong overall performance, especially on the challenging industry task.

Figure 4 presents the confusion matrices computed using the chunking model, which offers multi-chunk representations suitable for fine-grained error analysis while also reducing the computational cost of generating these visualizations. Figure 4a shows that gender prediction is highly reliable, with errors affecting only about 8% of the test set. Age-group classification(Figure 4b) exhibits clear diagonal dominance across all three categories. The 18–29 group achieves the highest recall, while the 13–17 group shows the least cross-class confusion. Most remaining errors arise between the two adult brackets (18–29 and 30–48), with confusion distributed roughly symmetrically in both directions. Industry performance(Figure 4c) varies substantially by class frequency. Smaller industries, particularly Finance & Property and Non-Profit, which together account for only about 7% of the dataset, exhibit negligible diagonal mass and are never predicted correctly. In contrast, high-support categories such as Student, Technology, and Education (collectively ~32% of the data) exhibit strong within-class accuracy. These patterns reveal strong performance on well-represented classes but persistent challenges on low-support or stylistically overlapping categories.

Figure 5 shows UMAP projections of the [CLS] representations learned by our model. When colored by gender (Figure 5a), the embedding space splits into two remarkably clean, compact clusters with virtually no overlap, consistent with the observed gender F1 of 0.90 and confirming that robust stylistic cues are captured even in a multitask setting. Age groups (Figure 5b) form three well-defined regions: the dominant 18–29 cohort occupies a dense central area, while the younger (13–17) and older (30–48) groups appear as distinct arms with only minor mixing at the adult boundary. The 14-class industry projection (Figure 5c) reveals a more complex yet highly structured manifold: major industries (e.g., Education, Technology, Arts) correspond to large, coherent clusters, whereas rarer categories are more scattered and partially merge

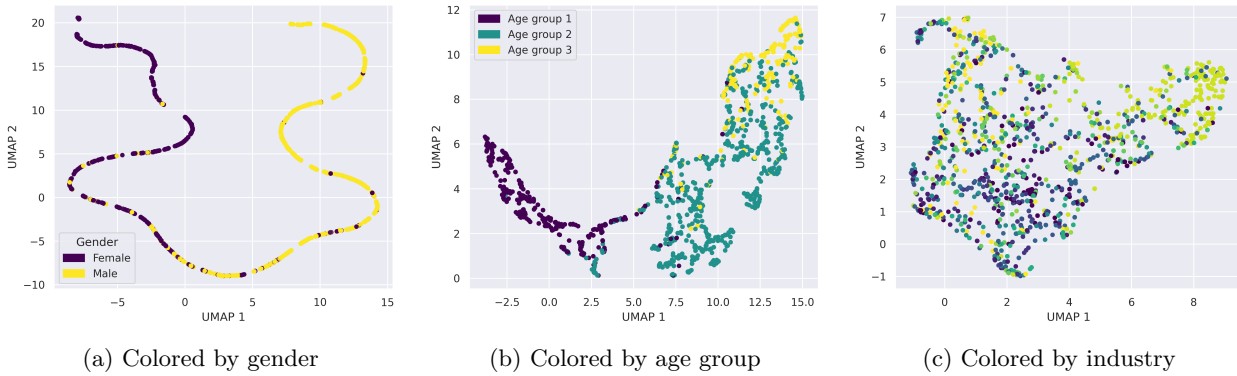

(a) Colored by gender      (b) Colored by age group      (c) Colored by industry

Figure 5: UMAP 2D projections of the final embeddings produced by the DeBERTa (chunked) model, colored by different task labels.

with semantically similar groups. This pattern directly explains the moderate macro-F1 of 0.24 on the industry task and underscores the benefit of longer contiguous context for disambiguating subtle topical and professional signals.

## 5 Discussion

Our study shows that multitask learning improves author-profiling performance across gender, age, and industry, including an 11% macro-F1 gain for industry over strong single-task baselines. Models with broader input context perform best overall, suggesting that long-form blog posts benefit from representations that capture extended stylistic and topical cues. Our comparison of long-document strategies indicates that different approaches highlight distinct aspects of the text and thus offer complementary advantages. Error patterns and embedding visualizations further reveal the relative difficulty of age and industry prediction due to overlapping semantic cues and boundary ambiguities. Overall, the results demonstrate that robust demographic inference on long-form blog text requires effective multitask representations and careful handling of document length.

Multitask learning succeeds because gender, age, and industry share overlapping lexical and topical cues, enabling richer representations than single-task training. Longer context helps for the same reason: diagnostic signals are sparse and distributed across posts. In contrast, chunking fragments the document, disrupting global structure and raising computational cost, while summarization compresses the text, often removing fine-grained cues and subtle demographic signals needed for accurate prediction.

To assess the sensitivity of the industry loss weight, we compared weights of 1.0, 1.5, and 2.0 during pilot training. Increasing the weight from 1.0 to 1.5 not only improves industry F1 substantially but also yields small gains on gender and age prediction, suggesting that industry prediction acts as a rich auxiliary task that enriches the shared encoder representations and benefits all tasks jointly. However, increasing the weight further to 2.0 leads to performance degradation across all tasks, indicating diminishing returns beyond the chosen value.

The relatively modest industry classification performance reflects two compounding factors. First, several industry categories exhibit substantial semantic overlap: Student and Education share similar topical and lexical signals, Technology, Science & Technical, and Internet overlap considerably in vocabulary and writing style, and Arts, Creative Media & Culture, and Communications-Media occupy adjacent semantic spaces. These boundaries are challenging to distinguish, as professional identity in blog writing does not always manifest through distinct linguistic markers. Second, class imbalance compounds this challenge — the largest category (Student, 11.12%) contains approximately four times more instances than the smallest (Finance & Property, 2.74%), leaving low-support categories with insufficient training signal for reliable prediction. Importantly, this imbalance is not an artifact of our dataset construction but reflects the natural distribution of professional backgrounds in the real world. Any author profiling system deployed in practice

will encounter similar skews, and our results therefore offer a realistic indication of expected performance under real-world conditions.

The confusion matrix Figure 4 reveals that several industry categories are never predicted correctly, with the model defaulting to high-frequency classes such as Student and Education. This pattern is partly attributable to the chunking-based representation used for this analysis: by segmenting documents into short independent chunks, global professional context is fragmented, leaving the model without the long-range topical signals necessary to distinguish low-support categories. This is consistent with our finding that DeBERTa-500, which processes full 500-token contiguous sequences, achieves much higher industry F1 (0.55) compared to the chunked variant (0.24). The confusion matrix thus illustrates not only the class imbalance challenge but also the critical importance of preserving document-level context for reliable industry inference — a finding that reinforces our recommendation against chunk-based processing for professional attribute prediction.

Our MTL framework achieves +11 points over the strongest single-task baseline on industry prediction — a task that prior models struggled with. Our results not only demonstrate strong performance on this challenging task but also provide actionable guidance for researchers approaching author profiling and demographic inference from long-form text, offering an empirical guidance on what works and what does not. More broadly, our finding that different prediction targets require different context ranges can be applicable to other multitask classification settings involving long documents, extending the relevance of this work beyond author profiling.

## 6 Conclusion

We introduce the first transformer-based multitask framework that jointly predicts gender, age group, and 14 industries from long-form blog text, provide the first systematic evaluation of input-length strategies for this setting, and offer new evidence on how demographic signals depend on context range, clarifying why gender is local, age is medium-range, and industry requires global topical coherence, supported by embedding-space analyses that reveal how shared encoders organize these signals.

The findings of this study have implications for the design of real-world author profiling systems. Gender and age-group prediction show consistent reliability across input strategies, suggesting these attributes are suitable for deployment in downstream applications. Industry classification, however, requires careful consideration — professional cues are sparse and globally distributed, and low-support categories remain challenging even under optimal conditions. Any system that infers demographic attributes from text must account for the ethical dimensions of such inference, including potential privacy concerns and the risk of unfair treatment of underrepresented groups. We recommend that deployed systems incorporate human oversight for high-stakes decisions and treat model predictions as probabilistic signals rather than definitive labels.

Future work may deepen bias analysis by examining how stylistic cues align with demographic attributes and by modeling the latent stylistic dimensions that drive these associations. It would also be valuable to incorporate insights from recent fairness-aware multitask learning methods and bias-auditing frameworks, given the sensitive nature of demographic inference. Ensuring the robustness and generalizability of the model across different domains should also be addressed. A key limitation of current author profiling research—including our own, is the assumption that writing style is static. In reality, individuals systematically shift their vocabulary, syntax, and topical focus across time, audience, and life stage, producing substantial intra-author variation. Our embedding analyses already hint at this dynamism: age-cohort clusters stretch and overlap in ways that reflect gradual linguistic evolution rather than fixed traits. Developing models that explicitly account for such personal evolution—whether through temporal modeling, continual learning, or diachronic multitask objectives—represents a critical and exciting direction for achieving truly robust and temporally stable profiling systems. The structure visible in our present data suggests that this goal is not only necessary but entirely feasible.

**Acknowledgments**

This research was funded by the Ministry of Science and Higher Education of The Republic of Kazakhstan within the framework of project AP23487777.

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
