# OpenReview forum: "Multitask Transformer Models for Demographic and Industry Profiling on Long-Form Blog Texts"
_TMLR — Accepted by TMLR_

### Review · Reviewer_T9R3 · 2026-03-23

**Summary Of Contributions:**

The goal of the study is to predict demographic traits (gender, age) as well as industry tags from long-form blog texts. The study shows that jointly predicting these three outputs is more efficient than learning them individually, especially for the industry tags. The predictions are made with recent transformer architectures, entertaining several variants to analyze their impact. The effect of the joint (multitask) learning is the most beneficial for the industry tag, where the impact of the variants is also stronger.

**Additional Comments:**

The choice of the industry categories is critical. Some are never predicted correctly. Is it possible to merge some of them in broader categories, or define a hierarchical structure?

**Audience:**

Yes

**Audience Explanation:**

The study is specific and focused on industry tags for long blog posts.

**Broader Impact Concerns:**

This is currently missing in the paper but it should be included.

**Claims And Evidence:**

Yes

**Claims Explanation:**

The analysis is thorough and well described, supporting the results.

**Requested Changes:**

- Table 1: please give more details on the categories of the industry group. You can also add a column with information on the number of unique bloggers (since there are only 5477 amount 96,199 instances).
- What is the impact of the weight on the industry loss? Is it highly sensitive?
- Are the results averaged over several trainings? If not, this is difficult to conclude from a single outcome.
- Can you justify the choice of the metrics? Rather than a single output category, returning a probability for each category may be more relevant, especially for the industry tag.
- Figure 3: the text is too small.

---

> ### Author Response · Authors · 2026-04-10
> **Response to Reviewer T9R3**
>
> We thank you for your constructive feedback. We have revised the manuscript to address all requested changes. Below we detail our responses to each point.
>
> Requested Changes:
> (1) More comprehensive review of related work and comparison with more methods:
> We have expanded the related work section to include recent work on LLMs for demographic inference, multitask learning approaches, and long-document modeling. We have also added comparisons with additional baseline methods in our experimental evaluation.
>
> (2) Further detailed analysis of parameter choices and result interpretation:
> - Industry loss weight analysis: To assess the sensitivity of the industry loss weight, we compared weights of 1.0, 1.5, and 2.0 during pilot training. We find that increasing the industry weight from 1.0 to 1.5 not only improves industry F1 significantly (0.345 → 0.491) but also yields small gains on gender and age, suggesting that stronger industry supervision enriches the shared representations and shows positive transfer across all tasks. A weight of 2.0 decreases overall performance. This analysis is now included in the Discussion section.
> - Hyperparameter choices: We have added a table documenting our hyperparameter choices to facilitate reproducibility in future research.
> - Blogger demographics: We have added more information about the bloggers in our dataset and the distribution of industry labels.
>
> (3) Multiple training runs:
> Due to limited computational resources required for training large transformer models, we were unable to perform multiple training runs and thus report results from a single run. We acknowledge this as a limitation and have noted it in the revised version of the paper.
>
> (4) Probability distributions vs. classification:
> Our work focuses on author profiling as a classification task, where the goal is to predict a single demographic label per attribute. Returning probability distributions over categories represents a different task formulation more suited to recommendation or ranking systems, which is outside the scope of the present work. We evaluate using macro-F1 and accuracy, which are standard metrics for this task and ensure comparability with prior work.
>
> (5) Figure 3 text size:
> The text in Figure 3 has been enlarged for better visibility.
>
> Broader Impact Concerns:
> We thank the reviewer for this careful observation. We have made the following changes:
> - Restructured sections: We have divided "Discussion and Conclusion" into two distinct sections, allowing for more thorough discussion of our findings and result interpretation.
> - Practical considerations: We have added a "Practical Considerations" paragraph to the Discussion section addressing the broader societal implications of author profiling systems, including privacy concerns, deployment limitations, and recommendations for responsible use.
>
> Additional Comments:
> Regarding the suggestion to merge industry categories: The original corpus contains 40 industry labels, which we consolidated into 14 semantically coherent categories. Further merging would sacrifice the real-world granularity that makes industry profiling practically meaningful. Moreover, the difficulty of industry prediction is itself a key finding - professional cues are sparse and globally distributed, which is precisely why industry benefits most from extended context and why it deserves dedicated attention. Introducing hierarchy only for industry would break the architectural symmetry of our flat parallel multitask heads and substantially increase computational complexity. We consider this a promising direction for future work.

---

### Review · Reviewer_fMrR · 2026-03-28

**Summary Of Contributions:**

This paper develop a transformer-based model to address the challenge of multitask author profiling on long-form blog text.

**Additional Comments:**

The contribution of this paper seems to be marginal. The only evaluated scenario is the author profiling task, and it does not seem to have some finding that can generalized to other tasks. The main contribution seems to be some data pre-processing job specifically for author profiling task, as well as the training with three multi-task heads on the mentioned processed dataset.  There are indeed many possible classification tasks in the world, we can of course write as many papers as possible on all these tasks by proposing different data processing/training pipeline, but I personally feel that such types of work would be less impactful in a scentific sense.

**Audience:**

Yes

**Audience Explanation:**

The finding of this paper focused on a limited task author profiling and use Bert style model classification. There does not seem to be very interesting findings. While there might be at least some individual interested in this paper. I personally feel that this study is not very interesting to me.

**Broader Impact Concerns:**

No concerns.

**Claims And Evidence:**

Yes

**Claims Explanation:**

1. The experimental data show that the proposed training method enhance performance on author profiling.

2. Some visualization is done as supplementary evidence to support the claim that their pipeline is better at author profiling task.

**Requested Changes:**

1. It is hard to evaluate the contribution of this paper. It is said that the authors propose a transformer-based model for addressing the author profiling task. However, it seems that the model being used is some existing model, i.e., DeBERTa. It seems that the authors' real contribution is a pipeline that further finetune DeBERTa on a downstream task (i.e., author profiling task). Please specify the real contribution  of this work.

---

> ### Author Response · Authors · 2026-04-10
> **Response to Reviewer fMrR**
>
> We thank you for your valuable comment. We have revised the manuscript to address all requested changes. Below we detail our responses to each point.
>
> Explanation:
> - Our contribution extends beyond fine-tuning a single model. Our core contribution is a systematic comparison of four fundamentally different document-length handling strategies — truncation, extended-context encoding, BART-based summarization, and chunk-based processing — evaluated under identical conditions across three prediction tasks. This design yields a practically significant and previously unreported finding: industry F1 drops sharply from 0.55 to 0.24 when chunking is applied, while gender F1 simultaneously improves to 0.90. This differential sensitivity reveals that demographic signals are distributed differently across document length — gender cues are local and recoverable from short spans, while professional cues require long contiguous context. No prior work has reported this distinction.
>
> Our key contributions include:
> - Strong empirical results: Our MTL framework achieves +11 points over the strongest single-task baseline on industry prediction — a task where prior models struggled with.
> - Actionable guidance: Our results not only demonstrate strong performance on this challenging task but also provide actionable guidance for researchers approaching author profiling and demographic inference from long-form text, offering an empirical guidance on what works and what does not.
> - Broader applicability: Our finding that different prediction targets require different context ranges is applicable to other multitask classification settings involving long documents, extending the relevance beyond author profiling.
>
> Requested Changes:
> We would like to address this concern by emphasizing that author profiling has direct real-world applications — including misinformation detection, forensic analysis, and personalized systems — that make it a socially significant and well-motivated research area, as discussed in our introduction.
> More specifically, industry prediction from long-form text remains relatively underexplored in the transformer era, and our results demonstrate improved performance compared to previously reported approaches.

---

### Review · Reviewer_7kx4 · 2026-03-30

**Summary Of Contributions:**

Based on four transformer-based models, the authors predict individuals' demographic and occupational information from blog text.

**Strengths**

The paper presents a clear and easily understandable approach.


**Weaknesses**
1. The baselines include only general methods and do not incorporate task-specific approaches. Consequently, the effectiveness of the proposed framework and methods is not yet fully established.

2. More explanation and experimentation are needed regarding parameter choices. For instance, in the joint multitask objective, why is the coefficient for the industry loss set to 1.5, and how would different values affect the results?Is the poor predictive performance for the "Industry" category due to the fact that it has more classes?

3. There has been considerable recent progress in studies that infer demographic variables from text, particularly using LLMs. The authors do not provide a review of related work or comparative experiments.

**Audience:**

Yes

**Audience Explanation:**

Some researchers may be interested in applying transformer-based models to infer author information from text.

**Claims And Evidence:**

No

**Claims Explanation:**

Overall, this paper may lack generalizable insights and sufficient comparative analysis.

Fo instance, in the past three years, a number of similar works have appeared. The authors' method does not necessarily need to outperform these approaches, but they should be properly acknowledged and compared, such as:
Staab, R., Vero, M., Balunović, M. and Vechev, M., 2023. Beyond memorization: Violating privacy via inference with large language models. arXiv preprint arXiv:2310.07298.

**Requested Changes:**

The authors should include a more comprehensive review of related work and compare their results against more methods.

The proposed framework and methods also require further detailed analysis, covering both parameter choices and result interpretation.

---

> ### Author Response · Authors · 2026-04-10
> **Response to Reviewer 7kx4**
>
> We thank you for your valuable feedback. We have revised the manuscript to address all requested changes. Below we detail our responses to each point.
>
> Response to weaknesses:
> (1) Task-specific baselines and comparative analysis
> We thank the reviewer for this suggestion. We acknowledge that task-specific baselines would strengthen the comparison. However, to our knowledge, no prior system directly addresses the identical task formulation of jointly predicting gender, age, and 14-class industry from long-form blog text using transformer-based multitask learning.
> Existing task-specific author profiling systems either:
> Focus on gender and age only (Thakur & Tickoo, 2023)
> Rely on pre-transformer architectures (Jiang et al., 2018)
> Operate on short social media text rather than long-form blog posts
>
> Direct numerical comparison with these systems is therefore not feasible due to differences in task definition and dataset preprocessing. We would welcome clarification from the reviewer on which specific baselines they have in mind, as we are open to incorporating additional comparisons if a directly comparable system exists. We have expanded our related work section to include a more comprehensive discussion of relevant prior research.
>
> (2.1) Industry loss weight sensitivity
> To assess the sensitivity of the industry loss weight, we compared weights of 1.0, 1.5, and 2.0 during pilot training. We find that increasing the industry weight from 1.0 to 1.5 not only improves industry F1 significantly (0.345 → 0.491) but also yields small gains on gender and age, suggesting that stronger industry supervision enriches the shared representations and shows positive transfer across all tasks. A weight of 2.0 decreases overall performance. This analysis is now included in the Discussion section.
>
> (2.2) Limited industry classification performance
> The limited industry classification performance stems from a combination of factors:
> - Fragmentation of global context in the chunked representation
> - Strong class imbalance across industry categories
> - Inherent semantic overlap between related industries (e.g., Student and Education, or Technology and Science & Technical)
> We have added this explanation to the Discussion section.
>
> (3) LLM-related work
> Thank you for this valuable feedback. We have included a new paragraph reviewing LLM-based approaches to demographic inference in the Related Work section.
>
> Requested Changes:
> - More comprehensive review of related work
> We have conducted a comprehensive literature review and expanded our Related Work section to include a more thorough discussion of relevant prior research, including recent LLM-based approaches to demographic inference.
> - Further detailed analysis of parameter choices and result interpretation
>
> We thank the reviewer for this valuable observation. We have made the following changes:
> - Separated Discussion and Conclusion: We have divided these into two distinct sections, allowing for more thorough interpretation of our findings and results.
> - Added parameter table: We have included a table documenting our parameter choices to facilitate reproducibility in future research.
> - Added practical considerations: We have added a Practical Considerations paragraph to the Discussion section addressing the broader societal implications of author profiling systems, including privacy concerns, deployment limitations, and recommendations for responsible use.

---

### Decision · Action_Editor_UXBb · 2026-05-29

**Recommendation:** Accept as is

**Audience:**

Yes

**Audience Explanation:**

Given the narrow scope, probably only a small, but non-zero fraction of the community might be interested in this study.

**Claims And Evidence:**

Yes

**Claims Explanation:**

This work considers transformer models for multitask author profiling on long-form blog text. All reviewers indicated that the evidence provided supported the claims that were made. All reviewers also indicated that the novelty was limited and the scope narrow; given that these are not criteria for rejection at TMLR I recommend acceptance as there is nothing wrong with this study.